# Geotechnical Deformation Distributed Measuring Technology Research Based on Parallel Spiral Sensing Line

**DOI:** 10.3390/s23187847

**Published:** 2023-09-13

**Authors:** Xinyu Miao, Qing Li, Renyuan Tong, Jun Wang, Chaopeng Li, Wenhao Tang

**Affiliations:** National and Local Joint Engineering Laboratories for Disaster Monitoring Technologies and Instruments, China Jiliang University, Hangzhou 310018, China; mxinxin777@163.com (X.M.); tongrenyuan@126.com (R.T.); 15268158940@163.com (J.W.); lcp352357@163.com (C.L.); 15821781970@163.com (W.T.)

**Keywords:** geological hazards, parallel helix, characteristic impedance, geotechnical deformation, time domain reflection technique

## Abstract

The precursors that appear when geological disasters occur are geotechnical deformations. This paper studies the TDR (Time Domain Reflection) measurement technology for the distributed measurement of geotechnical deformation using parallel spiral wire as a sensor, which is used for monitoring and early warning detection of geological disasters. Based on the mechanism of the electromagnetic field distribution parameters of the parallel spiral sensing wire, the relationship between the stretching amount of the parallel spiral wire and the change in its characteristic impedance is analyzed. When the parallel spiral wire is buried in the soil, the geotechnical deformation causes the parallel spiral wire to be stretched, and according to its characteristic impedance change, the stretching position and the stretching degree can be obtained, thus realizing the distributed measurement of geotechnical deformation. Based on this principle, the TDR measurement system is developed, and a local single-point stretching amount and stretching positioning experiment are designed for the parallel spiral sensing line to verify the effectiveness of the sensing technology and the usability of the measurement system.

## 1. Introduction

The earth, on which mankind depends, is not peaceful. Hundreds of geological disasters occur worldwide every year, and landslides, mudslides, and other types of geological disasters occur frequently in earthquake-prone areas [1]. In recent years, with human intervention and influence on nature, man-made geological disasters have increased year by year, such as tailings ponds that are prone to collapse and mountains where vegetation has been destroyed. Once a geological disaster occurs, it will pose a serious threat to the safety of human life and property [2]. For example, on 24 June 2017, a high-level landslide occurred in Xinma Village, Feixi Town, Maoxi County, Aba Prefecture, Sichuan Province, China, resulting in 10 deaths, 73 missing persons, and 405 emergency relocations, causing direct economic losses of up to 178 million yuan [3]; on 23 July 2019, a landslide occurred in the Fork Ditch Group of Pengdi Village, Jichang Town, Shuicheng County, Liupanshui City, Guizhou Province, China. The landslide caused 43 deaths and 9 missing people, and more than 700 other people were urgently relocated, with losses amounting to 190 million yuan [4]; in the same year, a landslide occurred in the Kavarapara region of India, causing 46 deaths and 11 missing people [5]. Despite the large amount of money invested in the prevention and control of geological hazards in the world every year, the threat of geological hazards to human life and property safety still exists, and how to realize effective monitoring and early warning of geological hazards has always been an urgent problem to be solved. To effectively monitor the hidden spots of geological hazards, many scholars around the world have proposed single or fused monitoring means based on various elements such as soil water content, subsurface deformation, surface deformation, rainfall intensity, and slope inclination angle. Geotechnical deformation, represented by subsurface deformation and surface deformation, is one of the most important physical quantities to be monitored when geological hazards occur.

The current measurement of geotechnical deformation in the world mainly includes total station observation technology [6,7,8], GPS navigation satellite positioning technology [9,10,11], remote sensing technology [12,13,14], pull-wire displacement sensing technology [15,16], fiber optic sensing technology [17,18,19,20], coaxial cable sensing technology [21,22,23], computer vision techniques and numerical simulation techniques [24,25], etc. The total station is a three-dimensional coordinate measurement system consisting of three main components: an optical distance meter, an electronic longitude and latitude meter, and an automatic data recording device. F. Glueer et al. [26] used total station technology to establish a measurement system for monitoring the rock slide movement of the Great Aletsch Glacier in the Swiss Alps. The advantage of this technology is that it continuously receives high-precision real-time monitoring data from remote locations and enables monitoring of geological hazards even in harsh environments where manual operations are not convenient, but the technology is point-distributed and is highly influenced by the weather. GPS technology was developed early, and in 1995, H Kondo and ME Cannon [27] developed a surface displacement monitoring system for landslide measurement using GPS technology that can detect about 2 cm of displacement; this technology has a large monitoring range, is not affected by the weather, and collects information in real-time and efficiently; however, it also belongs to point distribution measurement, which cannot realize distributed measurement and is constrained by the good or bad satellite signal. Like the above two techniques, remote sensing does not require direct contact with the measured target, and the acquisition of geometric and physical information about the target feature is achieved by using sensors to collect and receive electromagnetic wave information reflected or emitted from the object. G Xu et al. [28] used remote sensing to monitor landslides and mudslides in the Loess Plateau region of China, but this technique, when used outdoors, is easily obscured by the vegetation on the rock and soil surface, which seriously affects the measurement accuracy. Pull-wire displacement sensing technology is more accurate than the previous three technologies; the resolution can reach 0.01 mm. J. Corominas et al. [29] used this technology to monitor landslide displacement in the Eastern Pyrenees and the Dolomites. The technology can also only measure the relative displacement of a few points, cannot achieve distributed measurement, and is prone to measurement where the change has not occurred; the location where the change occurred cannot be measured. Fiber optic sensing technology has good results for the measurement of smaller deformations, but its application is limited to scenarios with large displacement deformations, such as landslides. YJ Sun et al. [30] used the technology for monitoring the deformation of landslides to overcome the defects of traditional sensors. The coaxial cable sensing technique is based on coaxial cable as a sensor combined with the time-domain reflection method. Yunmin Chen et al. [31] used this method to measure soil shear displacement because of the limited amount of coaxial cable stretching, so only the longitudinal shear deformation can be measured. Meanwhile, with the rapid development of computer science, computer vision technology and numerical simulation technology play an increasingly important role in monitoring and early warning analysis of geotechnical deformation. Canming Yuan et al. [32] used the slopes in Dayu County, China, as a research background to establish a test model and proposed a new automatic identification method based on depth information, i.e., monitoring and analyzing landslide deformation areas by comparing the depth information of the initial point cloud and the landslide point cloud while combining with the k-means clustering algorithm. However, this method has a limited monitoring distance and is based on the vulnerability of computer vision to interference from natural factors such as rainfall.

Aiming at the shortcomings of the above methods, such as low accuracy, great influence by the environment, and a limited stretching amount, combined with the features that a parallel helix can stretch a large amount and can measure lateral and longitudinal geotechnical deformation at the same time, this paper proposes a low-cost and high-precision measurement method based on the combination of a time-domain reflectance method with the parallel helix as a distributed sensor to realize the distributed measurement of large deformation scenarios, such as landslides and subsidence, to study the mechanism of the tensile deformation, and to develop the measurement device. This technology can be used in locations where there is a potential for geotechnical deformation, such as mountain cliffs, slopes prone to mudslides, tailing ponds, and so on.

Challenges: First, there are many measurement methods for geotechnical deformation, but there are fewer studies on distributed measurement methods for geotechnical deformation over a wide range. Second, the exact relationship between the tensile deformation of a parallel helix and the change in its characteristic impedance is not well understood. Third, rapid reflection and high pulse energy loss during the process are significant features of the time-domain reflection technique, and the measurement device needs to be able to emit narrow pulses with high amplitude and ultra-bandwidth so that the reflected signals can be efficiently sampled, so this study requires a very high level of device design.

Contributions:For the first time, we apply the spiral sensing line and TDR technology in the direction of monitoring and warning of landslide disasters, and this study fills the research gap of large deformation distributed measurement in monitoring and warning of landslide disasters.We investigated the relationship between the tensile deformation of the parallel helix and the variation of the characteristic impedance and established a solution model for the distribution parameters of the parallel helix, thus obtaining the theoretical relationship between the tensile deformation of the parallel helix and the variation of the characteristic impedance.We designed the corresponding measuring device and parallel helix stretching experimental platform for the theoretical method proposed in this paper, which can effectively simulate the occurrence of geologic disasters for experiments.

The paper is organized as follows: The basic structure of the sensing element parallel helix in this paper is presented in Section 2. Section 3 starts by introducing the structure of the parallel spiral sensing line, then analyzing its sensing mechanism, establishing the solution model of distributed parameters, studying the relationship between its stretching case and the change of characteristic impedance, and introducing the principle of the time-domain reflection technique. Section 4 describes the construction of the measurement system. Section 5 obtains experimental data by setting up a tensile measurement experimental platform and using the device to perform tensile localization experiments and tensile shape variation experiments based on the change in characteristic impedance of the parallel spiral sensing line. Furthermore, the conclusions in Section 6 are that the experimental results demonstrate the effectiveness and usability of the parallel spiral wire for geotechnical deformation distributed measurements.

## 2. Parallel Spiral Structure

The parallel spiral wire, which can also be called parallel spiral sensing wire according to its use, offers the possibility of distributed measurement of large deformations. It consists mainly of two parallel copper wires wrapped in a silicone outer skin, a central silicone strip, and a protective sleeve of outer silicone material [33].

Depending on the type, the two copper wires are twisted and wound from 40/60/100/150 pieces of 0.08 mm-diameter tinned copper wire. To ensure that the spacing between the two copper wires is small enough and consistent everywhere, the two copper wires are closely glued together in parallel, and then the two glued parallel copper wires *A* and *B* are spirally wound on the central silicone strip with the same spacing, and the unit length of the parallel spiral wire requires 5 times the length of the parallel copper wire to be spirally wound and finally glued to form the outer protective sleeve. Since the silicone material has good elasticity, the parallel spiral sensing line has the feature of a large stretchable amount. The actual parallel spiral sensing wire is shown in Figure 1a, the structure schematic is shown in Figure 1b, and the parallel copper wire structure is shown in Figure 1c.

Unlike the common coaxial wire, due to the more complex structure of the parallel spiral wire, the current research on its sensing mechanism and deformation measurement is not deep enough, lacking sufficient theoretical analysis and accurate mathematical models, and the theoretical analysis needs to include the analysis of the interaction of its neighboring spiral wires, leading to its limited use. Therefore, in-depth analysis and study of the sensing principle of parallel spiral wires from the perspective of electric circuits and electromagnetic fields are necessary [34].

## 3. Parallel Helix Distributed Measurement Principle

### 3.1. Parallel Helix Sensing Principle Analysis

Parallel spirals exist as sensor elements in the distributed measurement method for large deformations described in this paper, which are laid along the geotechnical surface of the site to be measured [35]. If a geological hazard does not occur, the structure of the parallel spiral remains unchanged and its corresponding parameters remain unchanged so that the characteristic impedance of the sensing line is equal everywhere and does not change; once a geological hazard occurs and the geotechnical soil is deformed, the parallel spiral sensing line is stretched and deformed, and the structure changes, which leads to a change in the distribution parameters and a change in the characteristic impedance at the deformation, and the change in the characteristic impedance. The position and the amount of change of the characteristic impedance can reflect the location of the stretching of the parallel spiral and the size of the stretching, which corresponds to the location and the degree of geotechnical deformation. In this way, the sensing between the appearance of the geotechnical deformation phenomenon and the measurable change of the characteristic impedance is realized. The parallel helix distributed measurement sensing principle is shown in Figure 2. To obtain the relationship between the stretching deformation of the parallel helix and the change of characteristic impedance, this paper analyzes the circuit and electromagnetic field, establishes the solution model of the distribution parameters of the parallel helix, finds the distributed capacitance and distributed inductance, and then analyzes the theoretical relationship between the stretching deformation of the parallel helix and the characteristic impedance.

Since the frequency of the pulse signal is high in the time-domain reflection technique, the parallel spiral does not meet the requirement that the length is much smaller than the wavelength, so it belongs to the long-line transmission, which can be described by using a distribution parameter network to build a circuit model, taking the differential length Δ*x* on a section of transmission line, whose equivalent circuit is shown in Figure 3, and it can be described by using four distribution parameters: distribution resistance *R*, distribution inductance *L*, distribution conductance *G*, and distribution capacitance *C*.

The relationship between the characteristic impedance *Z* of the parallel helix and the distribution parameters *R*, *L*, *G*, and *C* can be obtained in the limit case when Δ*x* tends to 0, according to Kirchhoff’s voltage law and Kirchhoff’s current law [36]. In the measurement method discussed in this paper, the characteristic impedance is the most critical parameter.
(1)$$Z=\sqrt{\frac{R+j\omega L}{G+j\omega C}}.$$

Since the parallel helix is approximately a uniform transmission line, it is a low-loss transmission line (*R* << *jwL*, *G* << *jwC*) and can theoretically be regarded as a loss-free transmission line. At this time, *R* = 0, *G* = 0, the characteristic impedance can be expressed as:(2)$$Z\approx \sqrt{\frac{L}{C}}.$$

To obtain the relationship between the distributed capacitance *C* and the amount of parallel helix stretching, a section of the region is now intercepted along the parallel helix transmission direction, and its profile model is shown in Figure 4.

In Figure 4, *A* and *B* represent the two parallel and closely adhered copper wires in the parallel spiral, and three turns of the parallel spiral are selected for this figure, with the subscripts *R* and *L* indicating the right and left of the middle turn, respectively, and the subscript numbers representing the number of left and right parallel spiral wires. The equivalent radius *r* is 0.253 mm, the resistivity of the material *ρ* is 0.01851 Ω·mm^2^/m, and the distance between the two closely adhered parallel copper wires is a fixed value of d = 1.6 mm; the relative dielectric constant *ε* of the central silicone strip is 5.5 and 4.6 mm in diameter *D*; the angle *θ* between the wind direction and the vertical direction of the parallel spiral wire is 30° when it is not stretched. For parallel spirals, the conductance *p* is equal to *d* + *s*, and *s* is the spacing between two turns of parallel copper wire. Since the parallel copper wire is tightly wound on the central silicone strip, the initial value of *s* is equal to *d* in the unstretched state; in the stretched state, *s* changes with the intensity of stretching, and the conductivity also changes. Ideally, all other parameters of the parallel helix do not change with the stretching phenomenon. Figure 4a shows the profile model of the parallel helix when it is not stretched; Figure 4b shows the profile model of the parallel helix in the center of the stretched state, in which the derivative is not consistent everywhere on the whole line, the middle is the largest stretched area, and the normal area is reached by the left and right asymptotic areas.

When the electric signal is passed in, both parallel copper wires can be regarded as uniformly charged wires with charge densities of −*η* and *η*, respectively, and any point *P* is located at a distance *x* from *B*_0_ length between the stretching region *B*_0_ and *A_R_*_1_. According to the Biot-Savard theorem, the electric field direction is in the direction of the diameter vector perpendicular to the axis of the parallel helix, and the electric field strength *E_B_*_0*B*′0_ between *B*_0_ and *B*′_0_ is:(3)$$E_{B0B\prime 0}=\frac{\eta }{2\pi \varepsilon }\cdot \frac{L_{P}}{x\sqrt{x^{2}+L_{P}{}^{2}}}.$$
in (3), *L_P_* is the distance from *B*_0_ to the point *P* in the direction perpendicular to the axis, *L_P_* = *D*/2cos*θ*′, and the angle between the parallel helix winding direction and the vertical direction changes in the stretched state; *θ*′ is different from *θ* and needs to be measured.

According to the superposition theorem and Gauss’s theorem, the electric field intensity generated by the parallel spiral in the stretched state in Figure 4b at point *P* along the direction shown satisfies:(4)$$E=E_{AL}+E_{BR}+E_{BL}+E_{AR}.$$
in (4), the right subscript letters *A* and *B* on the right side of the equation represent the two parallel copper wires in Figure 3, and *L* and *R* represent the left or right side of the point *P*, respectively. According to (3), the four electric field strengths on the right side of the equal sign of (4) are solved as:(5a)$$E_{AL}=-\frac{\eta L_{P}}{2\pi \varepsilon }\sum_{i=0}^{K}\frac{1}{\left[i\left(s+d\right)+d+x\right]}\times \sqrt{\frac{1}{{\left[i\left(s+d\right)+d+x\right]}^{2}+L_{P}{}^{2}}},$$
(5b)$$E_{BL}=\frac{\eta L_{P}}{2\pi \varepsilon }\sum_{i=0}^{K}\frac{1}{\left[i\left(s+d\right)+x\right]}\times \sqrt{\frac{1}{{\left[i\left(s+d\right)+x\right]}^{2}+L_{P}{}^{2}}},$$
(5c)$$E_{AR}=\frac{\eta L_{P}}{2\pi \varepsilon }\sum_{i=0}^{K}\frac{1}{\left[i\left(s+d\right)+s-x\right]}\times \sqrt{\frac{1}{{\left[i\left(s+d\right)+s-x\right]}^{2}+L_{P}{}^{2}}},$$
(5d)$$E_{BR}=-\frac{\eta L_{P}}{2\pi \varepsilon }\sum_{i=0}^{K}\frac{1}{\left[\left(i+1\right)\left(s+d\right)-x\right]}\times \sqrt{\frac{1}{{\left[\left(i+1\right)\left(s+d\right)-x\right]}^{2}+L_{P}{}^{2}}}.$$
where *K* represents the number of all spirals on the left and right sides of the *P* point. The voltage value between *B*_0_ and *A_R_*_1_ can be known from Gauss’ theorem as:(6)$$U=\int_{r}^{s-r}Edx=\frac{\eta }{2\pi \varepsilon }\left(-\zeta_{AL}+\zeta_{BL}+\zeta_{AR}-\zeta_{BR}\right).$$

When the stretching phenomenon occurs, the parallel helix in a period is stretched, stretching the center of the degree of maximum, the more to the sides of the smaller, to facilitate the calculation, the approximation of *K* to take 3, then:(7a)$$\zeta_{AL}=\zeta_{BR}=\sum_{i=0}^{3}\ln\frac{\sqrt{{\left[\left(i+1\right)\left(s+d\right)-r\right]}^{2}+L_{P}{}^{2}}-L_{P}}{\sqrt{{\left[i\left(s+d\right)+d+r\right]}^{2}+L_{P}{}^{2}}-L_{P}}\times \frac{i\left(s+d\right)+d+r}{\left(i+1\right)\left(s+d\right)-r},$$
(7b)$$\zeta_{AR}=\zeta_{BL}=\sum_{i=0}^{3}\ln\frac{\sqrt{{\left[i\left(s+d\right)+s-r\right]}^{2}+L_{P}{}^{2}}-L_{P}}{\sqrt{{\left[i\left(s+d\right)+r\right]}^{2}+L_{P}{}^{2}}-L_{P}}\times \frac{i\left(s+d\right)+r}{i\left(s+d\right)+s-r}.$$
the distributed capacitance on the parallel helix is the ratio of charge density to voltage per unit length of wire A and B. Combining (6), it is obtained that:(8)$$C=\frac{Q}{U}=\frac{\eta }{U}=\frac{\pi \varepsilon }{\left(\zeta_{AR}-\zeta_{AL}\right)}.$$
due to the high conductivity of the parallel helix, the influence of the distribution resistance is theoretically insignificant. *μ* is the relative permeability of the material, in the medium of the intrinsic parameters (*μ*, *ε*):(9)LC=με.
substituting (8) into (9), the distributed inductance between the unit length of wires A and B is obtained as:(10)$$L=\frac{\mu }{2\pi }\cdot \zeta .$$
by relating (2), (8) and (10), the characteristic impedance expression of the parallel helix is solved as:(11)$$Z=\sqrt{\frac{\mu }{\varepsilon }}\cdot \frac{\zeta_{AR}-\zeta_{AL}}{\pi }.$$
after the derivation, it can be seen that the characteristic impedance of the parallel helix is only related to some of its structural parameters (*r*, *d*, *D*, *μ*, *ε*, *θ*′).

The structural characteristic parameters of the parallel helix are *r* = 0.253 mm, *d* = 1.6 mm, *D* = 4.6 mm, *μ* = 1, *ε* = 5.5, and *θ*′ ≈ 30°. Substituting these parameters into the characteristic impedance expression, the stretching amount of the parallel spiral wire is expressed in the change of the spacing *s* between the two turns of parallel copper wires, and the relationship between the characteristic impedance *Z* of the parallel spiral wire and the change of the stretching amount is obtained as shown in Figure 5. By observing Figure 5, the characteristic impedance of the parallel spiral line increases with the increase in stretching distance, and the trend of this increase is gradually slow. When *s* is certain and is located in the center of the stretching parallel spiral, the characteristic impedance is the largest, and characteristic impedance to both sides of the gradient region decreases until it is the normal unstretched region.

### 3.2. Principle of Time Domain Reflection Technology

The time domain reflection technique, often referred to as TDR, is commonly used for transmission line fault detection and signal integrity analysis, and it was used in the late 1980s to monitor the local deformation of rock masses [37]. The principle of TDR measurement is shown in Figure 6. The reflected signal is generated if the characteristic impedance is changed by stretching a position in the middle of the parallel spiral. The reflected signal is collected and transmitted to the upper computer using the equivalent sampling method, and then the relationship between the incident voltage and the reflected voltage is used to obtain the relationship between the change in the characteristic impedance so that the amount of stretching and the location of stretching can be obtained.

By using the transmission speed of the signal and the transmission time between the incident and reflected signals, it is possible to determine the location in the parallel helix where the characteristic impedance changes, i.e., where the stretching phenomenon occurs.
(12)$$L=\frac{v\cdot T}{2}.$$
(12), in which *l* is the relative position of the stretching deformation point from the incident end; *v* is the transmission speed of the signal, and the transmission speed of the pulsed signal in the copper medium is about 0.7 times the speed of light; and *T* is the time difference between the sampled incident signal and the reflected signal.
(13)$$\rho =\frac{V_{r}}{V_{i}}=\frac{Z-Z_{0}}{Z+Z_{0}},$$
*ρ* is the reflection coefficient, whose value is the ratio of the reflected signal voltage amplitude *V_r_* to the incident signal voltage amplitude *V_i_*. *Z* is the characteristic impedance of the parallel helix to be measured, and *Z*_0_ is the initial characteristic impedance (50 Ω) of the parallel helix when no tensile deformation occurs. (13) can be obtained at the stretching deformation of the parallel spiral line characteristic impedance and the relationship between the incident signal and the reflected signal:(14)$$Z=Z_{0}\cdot \frac{1+\rho }{1-\rho }=Z_{0}\cdot \frac{V_{r}}{2V_{i}-V_{r}}$$
(14) shows that the greater the reflected voltage, the greater the characteristic impedance at the stretching point of the parallel helix with a constant incident voltage. This chapter analyzes the connection between the stretching deformation of the parallel helix and its characteristic impedance change; that is, the stretching deformation of the parallel helix will lead to a change in the characteristic impedance of its stretching section. Based on this connection, the change in characteristic impedance at any point on the parallel helix can be obtained by observing the reflected voltage according to the TDR to obtain the stretching position and stretching amount of the parallel helix, which provides the theoretical basis for the research content of this paper.

## 4. Development of a Measurement System

### 4.1. Total System Solution

The hardware circuit of the system is the core part of the measurement system, and whether its design is reasonable directly determines the accuracy and effectiveness of the detection process. The block diagram of the overall scheme of the measurement system designed in this paper is shown in Figure 7, which includes three parts: a parallel helix, a hardware circuit part, and an upper computer. Among them, the hardware circuit can be divided into four main parts: a narrow pulse source module, an echo conditioning circuit module, an equivalent sampling circuit module, and an FPGA control module.

First, the FPGA control module generates the trigger signal of the narrow pulse source circuit to control the occurrence of narrow pulses, and the narrow pulse signal passes through the return conditioning circuit and also enters the parallel helix as the incident signal when it encounters the change of characteristic impedance, the reflected signal enters the return conditioning circuit, and the return conditioning circuit process the incident signal and the reflected signal in turn by first limiting the amplitude, then amplifying and then limiting the amplitude, and also by The FPGA also generates a timing control sampling circuit to sample the signal processed by the echo conditioning circuit, and finally transmits it to the upper computer to obtain data for analysis. The physical diagram of the circuit board is shown in Figure 8.

### 4.2. Narrow Pulse Source Design

The quality of the pulse signal generated by the narrow pulse source depends on the performance of the entire design. The length of the parallel spiral used in the experiment is 10 m, the equivalent transmission distance of the signal on the parallel spiral is 50 m, and the time from the signal into the incident end to the reflected signal at the end of the parallel spiral back into the incident end is only about 400 ns. Therefore, the rising and falling edges of the pulse signal should be narrow enough; otherwise, it will appear that the incident signal has not yet fully entered the incident end of the parallel helix, it has already generated the reflected signal, and the two signals are superimposed together by the system sampling, which cannot observe the obvious experimental phenomenon. At the same time, taking into account the energy loss in the signal transmission process, the amplitude of the pulse signal should be large enough [38]. This design uses an avalanche triode based on the Marx cascade circuit with a parallel circuit to propose a new parallel cascade structure for the circuit, and its basic working principle is shown in Figure 9. The energy storage element is first stored by the high-voltage power supply, and when the FPGA control module generates a trigger signal to control the fast switch formed by the avalanche triode to quickly close and then break, the energy storage element completes the instantaneous release of energy to form a pulse signal.

This pulse source circuit mitigates the trailing wave oscillation phenomenon caused by overshoot by connecting a 10 nF inductor in series with the load resistor [39]; changing all capacitors into the form of two small capacitors in parallel, which effectively shortens the pulse width without reducing the pulse amplitude; and reducing the capacitance value in the final parallel circuit, which is used to further shorten the rising edge of the pulse signal [40]. The new parallel cascade pulse circuit diagram is shown in Figure 10, and the narrow pulse signal generated by it is connected to the KEYSIGHT DSOS254A oscilloscope through a 20 dB attenuator, which has a real-time sampling rate of 2.5 GHz. A pulse signal with an actual amplitude of up to about 320 V and a pulse width of just 4.25 ns is obtained, and the actual narrow pulse image is shown in Figure 11.

### 4.3. Echo Conditioning Circuit Module Design

For the reason that the pulse signal generated by the narrow pulse source is too high in amplitude, it cannot be directly sampled. To prevent the pulse signal from causing irreparable harm to the device at the receiving end, the received signal should be limited to a certain voltage range by limiting the pulse signal to large peak-to-peak values. On the other hand, since the reflected signals at some slight stretching locations are relatively weak, direct acquisition is not convenient for observation, so they need to be amplified.

This circuit uses a multi-stage amplifier circuit combining a triode as an emitter follower and a common base amplifier circuit and is divided into three stages, with the front and rear stages being the emitter follower and the middle stage being the common base amplifier circuit. The emitter follower has a low output impedance, and the output signal is in phase with the input signal, which plays a role in isolating the front and rear stages, while the common base amplifier circuit has good frequency characteristics and is suitable for amplifying high-frequency signals. The final circuit will be limited once the signal output is limited to the equivalent sampling circuit. The echo conditioning circuit is shown in Figure 12.

### 4.4. Equivalent Sampling Module Design

Based on Nyquist’s sampling theorem, the sampling frequency needs to be more than twice the frequency of the sampled signal. The signals generated by the techniques in this paper are all high-frequency signals, and to obtain sampling results without distortion and with good waveform effects, the performance requirements of the ADC devices are particularly high. Considering the very good periodicity and repeatability of the pulse signal generated by the narrow pulse source controlled by the FPGA with a period of 10 us, the equivalent sampling method can be used [41], i.e., by sacrificing the sampling time to achieve the acquisition of high-frequency signals using a relatively low-speed ADC device, reducing the burden on the sampling module and subsequent data transmission.

The equivalent sampling method used in this paper belongs to the sequential equivalent sampling method, which uses the precise timing of the FPGA control module to precisely control the sampling signal, and the sequential equivalent sampling scheme is shown in Figure 13. The sampling clock of this scheme uses a high level of 20 ns, after which the low level is maintained until the next cycle is delayed by 4 ns relative to the previous sampling time. Since the whole process from the incident signal to the return of the reflected signal does not exceed 400 ns, when the 101st equivalent sampling is completed, the high level of the sampling clock is kept constant and the low level lasts 99.58 us, and then the above process is repeated, which is equivalent to collecting only the valid data within the first 400 ns of each cycle and discarding the useless data later, saving the acquisition time and data cache space. The equivalent sampling frequency of sequential equivalent sampling is 250 MHz, which meets the requirements of Nyquist’s sampling theorem. The hardware circuit uses a 12-bit precision AD9226 sampling chip, and the equivalent sampling hardware circuit is shown in Figure 14.

## 5. Experimental Platform and Measurement Verification

### 5.1. Construction of Experimental Platform

To verify the effectiveness of the parallel helix for the geotechnical deformation distributed measurement technique described in this paper, it is necessary to simulate the geotechnical deformation phenomenon for the parallel helix stretching experiment. While there are many unstable factors in simulating geotechnical catastrophes by burying them directly into the soil, it is not possible to determine the exact amount of stretching by directly stretching the parallel spirals manually. For the experiment, a parallel helix tensile test platform was built to simulate the parallel helix buried in the geotechnical soil, which can simulate the geotechnical deformation and deformation ranges, thus accurately controlling the tensile condition of the parallel helix. The experimental platform is shown in Figure 15. The platform is mainly composed of a stepping motor, a screw slide table, and a fixed flat plate. The parallel spiral line is fixed on the groove of the fixed flat plate by bolts, and there is a belt connector below the flat plate, which drives the flat plate to move when the screw slide table rotates. The rotation of the screw slide table is realized by the movement of the stepping motor. When geological disasters occur, the severity of geotechnical deformation is uncertain. Therefore, by controlling the stepping motor, the spiral sensing line can be stretched to a fixed length accurately to obtain the relationship between the different stretching degrees and the change in its characteristic impedance. At the same time, when geological disasters occur, the range of geotechnical deformation is also uncertain. The experimental platform can simulate different geotechnical deformation ranges by controlling different initial fixed positions of parallel spirals, and the parts outside the fixed areas will not be stretched.

### 5.2. Tensile Positioning Test

The distributed measurement experiments were carried out mainly by three parts: the tensile test platform and parallel helix, the system hardware circuit, and the upper computer. Relying on these three parts, a parallel helix with a length of 10 m and structural characteristics parameters of *r* = 0.253 mm, *d* = 1.6 mm, *D* = 4.6 mm, *μ* = 1, *ε* = 5.5, was selected. *θ*′ ≈ 30° of a parallel helix for the tensile positioning experiments of the parallel helix. The stretching points were located at 20-cm intervals starting from the end of the parallel spiral, for a total of 10 points. When stretching, the stretching points were selected to be fixed on the stretching platform at 15 cm each in front and behind and then stretched 2 cm in this position interval with the end of the parallel helix open. To eliminate abnormal data, 100 sets of data were collected at each stretching point, while the median was taken afterward, and the experimental results after superimposing the time-domain reflection curves of 10 points are shown in Figure 16.

According to the experimental results, the relative time of the reflected signal at each stretching point from the end of the parallel helix can be obtained, and by using (12), the theoretical stretching position of the experiment can be calculated. Table 1 shows the comparison results between the actual and theoretical values of the stretching position experiment, and the relative error is the absolute error between the actual and theoretical values and the percentage of the actual length of the parallel helix used in the experiment of 10 m. The results indicate that the technique can achieve a measurement accuracy of 0.1% for parallel helix stretch positioning. Based on the results of this experiment, it can be shown that in the practical use of monitoring geotechnical deformation, the specific location where the geotechnical deformation occurs can be determined based on the different locations of the reflected signals.

### 5.3. Local Single-Point Stretching Volume Test

In the same environment, the above-mentioned parallel spiral is still used for local single-point stretching experiments, and any point in the middle of the parallel spiral is chosen to be fixed on the experimental platform in a 15-cm interval before and after, and the stretching amount is increased from 0 mm to 60 mm at an interval of 5 mm. Similarly, 100 sets of data are collected for each stretching amount, the median is taken, and the results of 100 sets of experiments are superimposed after taking the median. After plotting as shown in Figure 17.

The reflection phenomenon on the right side of the figure is the result of the secondary reflection of the end reflection signal returning through the stretching position and does not affect the analysis of the whole experimental results. From Figure 17, it can be seen that the voltage of the reflected signal increases with the increase in the stretching amount, i.e., the characteristic impedance of the stretching point of the parallel helix increases with the increase in its stretching amount. The experimental measurement data for the local single-point tensile volume are shown in Table 2.

At the location of the stretching point, the voltage changes with the stretching degree, and the voltage at the stretching degree is subtracted from the voltage at the point when it is not stretched to get the reflected voltage at the stretching degree. It can be seen that the reflected voltage, the characteristic impedance, and the parallel helix stretching amount basically show a linear change, but with the increase in the stretching amount, the growth trend of the reflected voltage and the characteristic impedance gradually becomes slower, in line with the theoretical analysis in Section 3 of this paper.

## 6. Conclusions

In this paper, firstly, the sensing mechanism of parallel spiral wire is studied from the perspective of an electromagnetic field; the solution model of distribution parameters of parallel spiral wire is established; and the distribution capacitance, distribution inductance, and characteristic impedance are solved. Based on the theoretical analysis, the TDR technique of parallel spiral wire for geotechnical deformation distributed measurement is studied, and the measurement system is developed. The accuracy of using the TDR technique to measure the stretching position of parallel spirals was demonstrated by conducting experiments on the stretching position of parallel spirals. By conducting experiments on the local single-point stretching volume, it was concluded that the characteristic impedance of the stretching point of parallel spirals increases with its stretching volume, which is consistent with the theoretical analysis. It is further shown that the location and amount of stretching of parallel spirals can be detected by the TDR technique, which proves the effectiveness of the TDR technique of parallel spirals for geotechnical deformation distributed measurement. The experimental data are analyzed to find the relationship between the position and magnitude of the reflected voltage and the stretching position and the stretching amount, and this relationship is added to the host computer so that the stretching position and the stretching amount of the parallel helix can be displayed in real time, and an alarm is given when there is an abnormality to realize the function of real-time monitoring of geotechnical deformation of geological disasters. Furthermore, we plan to consider deeper analysis of the experimental sensor data in combination with numerical simulation techniques to extract more information on geotechnical deformation phenomena.

## Figures and Tables

**Figure 1 sensors-23-07847-f001:**
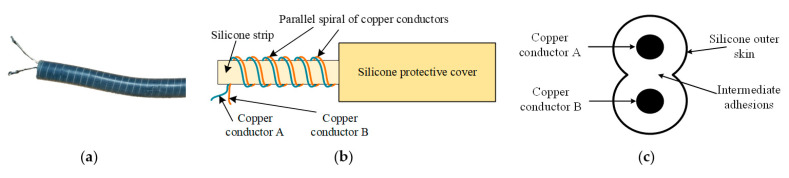
Parallel spiral sensing line. (**a**) Parallel spiral sensing line physical drawing; (**b**) Schematic diagram of parallel helix shape structure; (**c**) Schematic diagram of the internal structure of the parallel helix.

**Figure 2 sensors-23-07847-f002:**
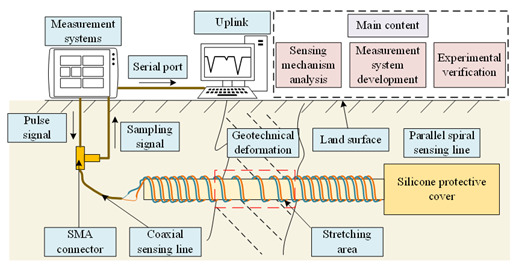
Parallel helix distributed measurement schematic.

**Figure 3 sensors-23-07847-f003:**
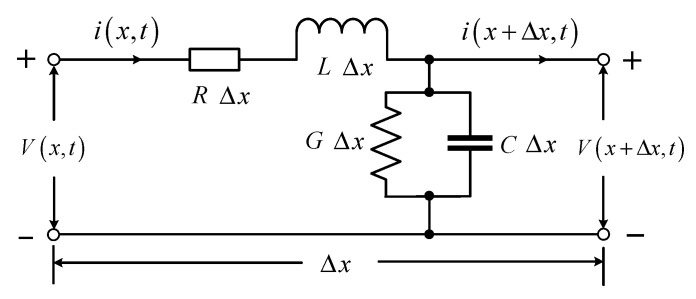
Parallel helix circuit model.

**Figure 4 sensors-23-07847-f004:**
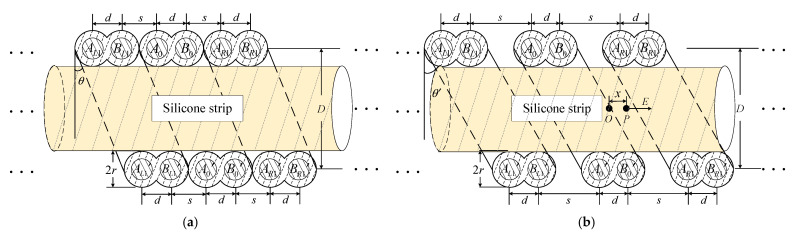
Parallel helix profile model. (**a**) The model before stretching; (**b**) Model after stretching.

**Figure 5 sensors-23-07847-f005:**
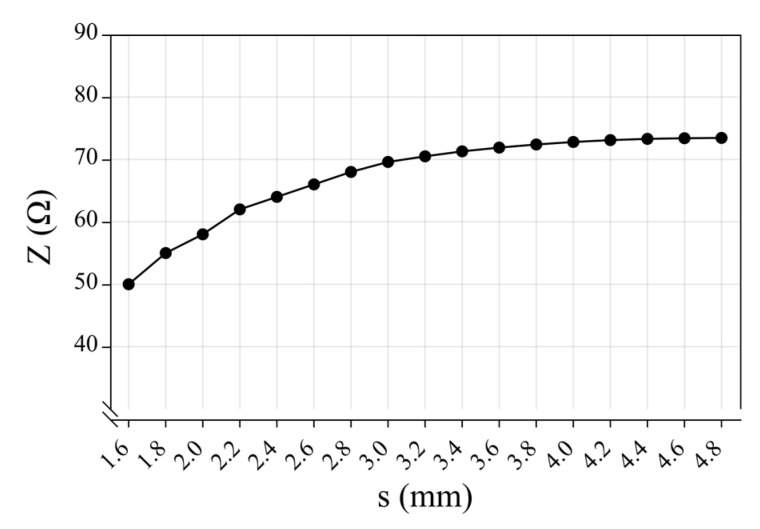
Parallel helix characteristic impedance curve with *s*.

**Figure 6 sensors-23-07847-f006:**
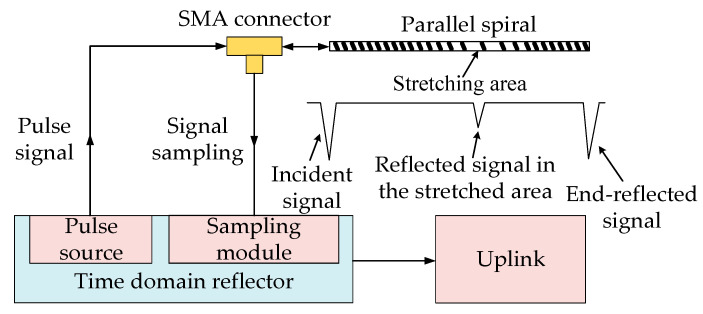
TDR measurement principle.

**Figure 7 sensors-23-07847-f007:**
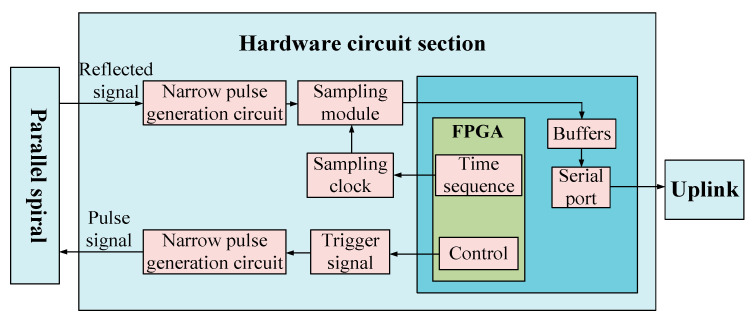
Overall solution diagram of the measurement system.

**Figure 8 sensors-23-07847-f008:**
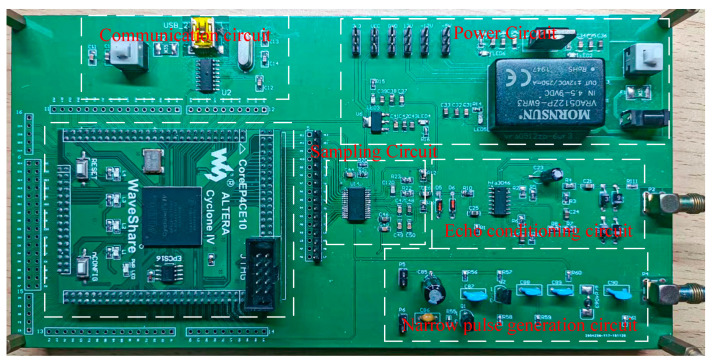
Physical drawing of the circuit board.

**Figure 9 sensors-23-07847-f009:**
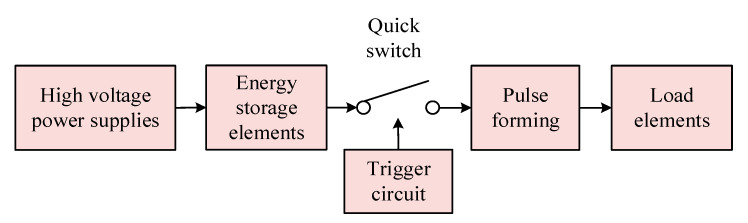
Working principle of pulse source.

**Figure 10 sensors-23-07847-f010:**
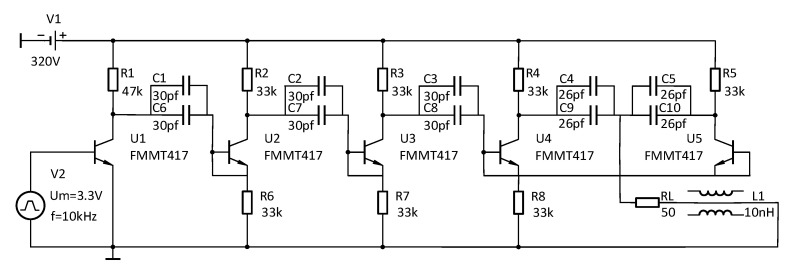
New parallel cascade circuit diagram.

**Figure 11 sensors-23-07847-f011:**
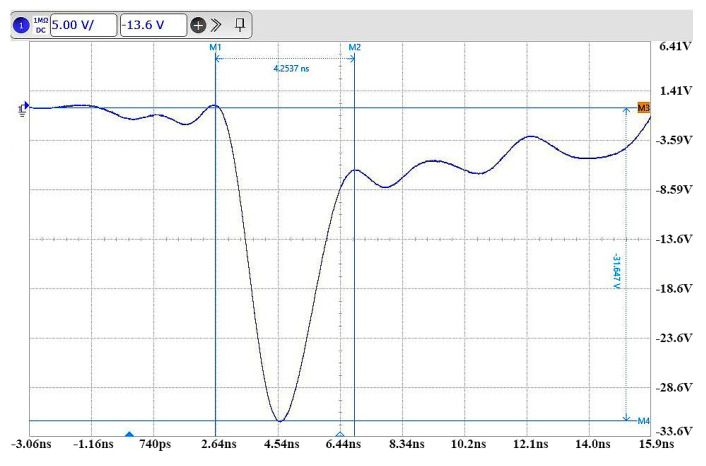
Actual narrow pulse signal diagram.

**Figure 12 sensors-23-07847-f012:**
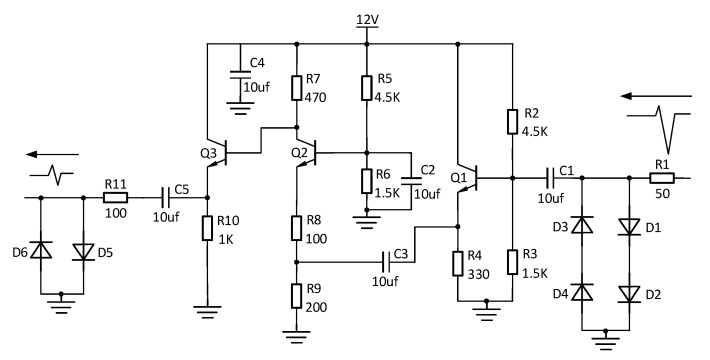
Echo conditioning circuit diagram.

**Figure 13 sensors-23-07847-f013:**
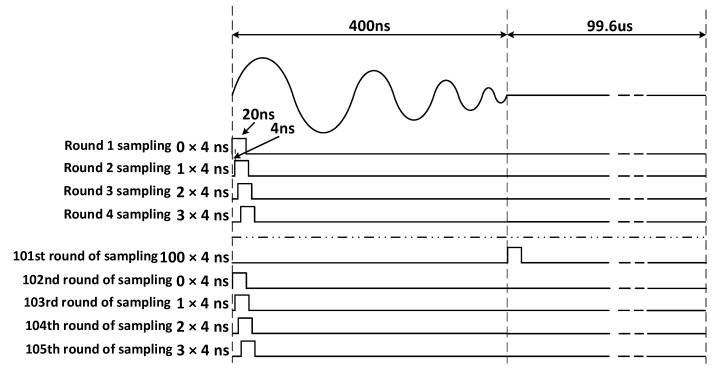
Sequential equivalence sampling scheme.

**Figure 14 sensors-23-07847-f014:**
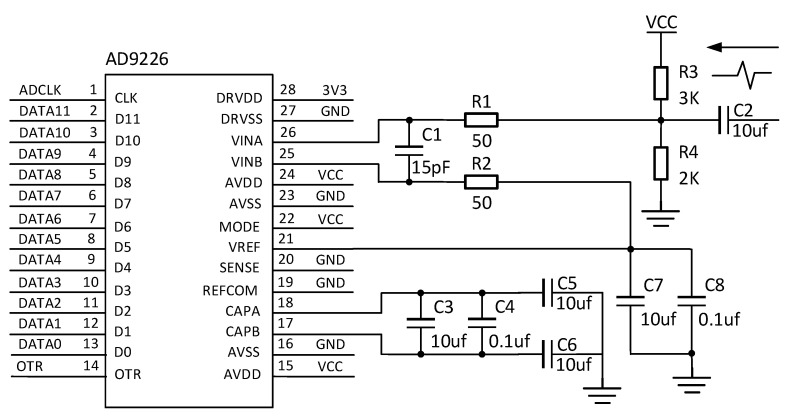
Equivalent sampling circuit.

**Figure 15 sensors-23-07847-f015:**
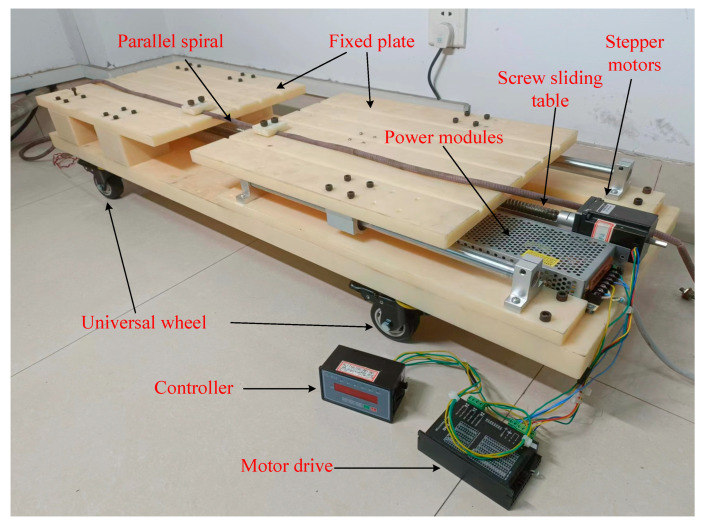
Parallel helix tensile test platform.

**Figure 16 sensors-23-07847-f016:**
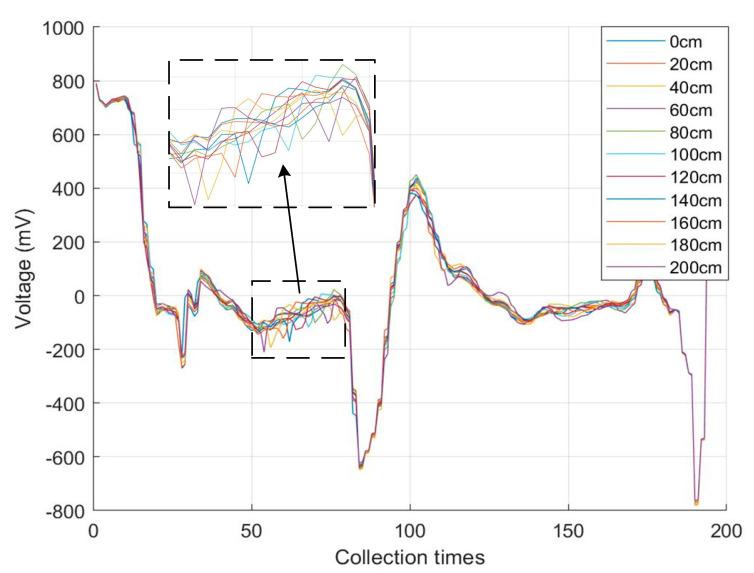
Tensile positioning test results.

**Figure 17 sensors-23-07847-f017:**
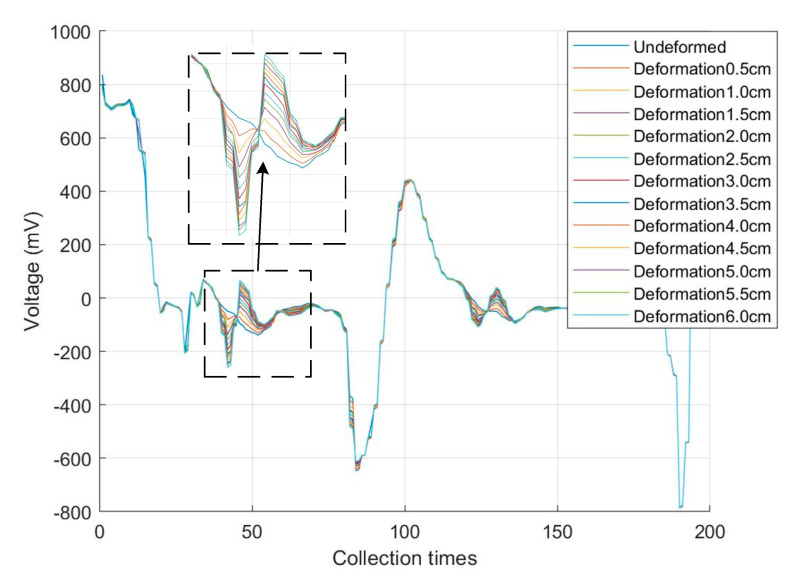
Experimental results of local single-point stretching volume.

**Table 1 sensors-23-07847-t001:** Tensile positioning measurement data.

Stretching Position (cm)	Measurement Results (cm)	Absolute Error (mm)	Relative Error (%)
0	0	0	0.00
20	20.3	3	0.03
40	39.7	3	0.03
60	60.1	1	0.01
80	80.5	5	0.05
100	100.6	6	0.06
120	119.9	1	0.01
140	140.0	0	0.00
160	160.3	3	0.03
180	180.5	5	0.05
200	199.7	3	0.03

**Table 2 sensors-23-07847-t002:** Local single-point stretch measurement data.

Tensile Strength (mm)	Voltage (mv)	Reflected Voltage (mv)	Characteristic Impedance (Ω)
0	−88.477	0.000	50.000
5	−120.215	31.745	53.530
10	−151.953	63.476	57.411
15	−175.391	86.914	60.497
20	−193.945	105.468	63.122
25	−209.570	121.093	65.340
30	−228.125	139.648	68.064
35	−240.820	152.343	70.192
40	−250.586	162.109	71.803
45	−261.328	172.851	73.609
50	−273.047	184.570	75.628
55	−281.836	193.359	77.226
60	−289.648	201.171	78.886

## Data Availability

Not applicable.

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
