# Peer review of "Geotechnical Deformation Distributed Measuring Technology Research Based on Parallel Spiral Sensing Line"

_sensors, 2023, doi:10.3390/s23187847_

Round 1

Reviewer 1 Report

I only have few minor suggestions to hopefully improve the quality of manuscript as below:

1. Line 9, what does TDR stand for, can you give the full name of the abbreviation?

2. In the “Introduction” section, the authors mentioned that geological disasters such as landslides and debris flows were of great harm, and the application of geotechnical measurement technologies, such as remote sensing and GPS, can prevent the occurrence of such disasters. However, with the rapid development of computer science in practical and physical scenarios, effective computer vision technologies and numerical simulation techno- logies can simultaneously carry out monitoring and early warning analysis. Therefore, the authors needs to supplement this kind of research in the “Introduction”. In addition, in view of the current development of technical research, I suggest the authors cite the following three relevant references:[1] https://www.sciencedirect.com/science/article/abs/pii/S0263224121014196 [2] [https://www.sciencedirect.com/science/article/abs/pii/S0263224123008527 [3] https://pubs.geoscienceworld.org/gsa/lithosphere/article/2021/Special%204/6426550/611090/Study-on-Three-Dimensional-Dynamic-Stability-of

3. Line 188, “according to Kirchhoff's voltage law and Kirchhoff's current law”these two laws need to be given relevant references.

4. Lines 223, 229, 238, 240,2 43, etc. Since the equation is preceded by a colon, the first letter of this part needs to be lowercase, for example, “In” is changed to “in” and “Due” is changed to “due”.

5. As shown in Figure 15, what is the innovation of the physics experiment carried out by the author? How to effectively solve the problem of this paper? Please add briefly in the manuscript.

6. Please add future research plans to this study. Will numerical simulation techniques be used with similar experiments or machine learning to make accurate predictions in the future?

The English language needs to be further modified and perfected

Reviewer 2 Report

The authors have presented a very unique sensing technology to detect geotechnical deformations based on parallel wire sensors and a TDR measurement technique. Overall, there are no questions regarding the technical rigor of this study and its laboratory demonstration. Certainly, the authors have acknowledged the possibility of improved theoretical development – which will be appreciated. All in all, I am happy with the current state of the manuscript, and it certainly warrants publication in sensors. However, I will appreciate it if the authors can shed some light on the following questions before this can be accepted for publication.

My main concern regarding this sensor stems from the question of the scalability of this sensor. It looks like this sensor requires a lot of invasive methods to be installed at the location of the monitoring site. It would be appreciated if the authors could discuss the possibility of their efforts in making this sensor more scalable and minimally invasive.

Having said that, can authors explain – on a comparative note – how their sensor is beneficial as compared to a more sensitive and minimally invasive fiber-based sensors such fiber bragg gratings?

It is also not clear whether how the authors propose to localize the geotechnical deformation site – e.g. if they consider such sensor is implemented in real scenario as the authors have identified in their introduction?

Overall, the English is well written throughout the paper. However, I would recommend the authors carefully proofread this manuscript again so as to detect some repeated words and sentence formations.
